# Antimicrobial Resistance Profile of Common Foodborne Pathogens Recovered from Livestock and Poultry in Bangladesh

**DOI:** 10.3390/antibiotics11111551

**Published:** 2022-11-04

**Authors:** Kazi Rafiq, Md Rafiqul Islam, Nure Alam Siddiky, Mohammed Abdus Samad, Sharmin Chowdhury, K. M. Mozaffor Hossain, Farzana Islam Rume, Md Khaled Hossain, ATM Mahbub-E-Elahi, Md Zulfekar Ali, Moizur Rahman, Mohammad Rohul Amin, Md Masuduzzaman, Sultan Ahmed, Nazmi Ara Rumi, Muhammad Tofazzal Hossain

**Affiliations:** 1Department of Pharmacology, Bangladesh Agricultural University, Mymensingh 2202, Bangladesh; 2Livestock Division, Bangladesh Agricultural Research Council (BARC), Dhaka 1215, Bangladesh; 3Antimicrobial Resistance Action Center, Bangladesh Livestock Research Institute, Dhaka 1341, Bangladesh; 4Department of Pathology and Parasitology, Faculty of Veterinary Medicine, Chattogram Veterinary and Animal Sciences University, Chattogram 4225, Bangladesh; 5Department of Veterinary and Animal Sciences, University of Rajshahi, Rajshahi 6205, Bangladesh; 6Department of Microbiology and Public Health, Faculty of Animal Science and Veterinary Medicine, Patuakhali Science and Technology University, Barisal 8210, Bangladesh; 7Department of Microbiology, Hajee Mohammad Danesh Science and Technology University, Dinajpur 5200, Bangladesh; 8Department of Microbiology and Immunology, Sylhet Agricultural University, Sylhet 3100, Bangladesh; 9Animal Health Research Division, Bangladesh Livestock Research Institute, Dhaka 1341, Bangladesh; 10Department of Physiology and Pharmacology, Faculty of Animal Science and Veterinary Medicine, Patuakhali Science and Technology University, Barisal 8210, Bangladesh; 11Department of Microbiology and Hygiene, Bangladesh Agriculture University, Mymensingh 2202, Bangladesh

**Keywords:** AMR, MDR bacteria, foodborne bacteria, animal origin food, Bangladesh

## Abstract

Multidrug-resistant (MDR) foodborne pathogens have created a great challenge to the supply and consumption of safe & healthy animal-source foods. The study was conducted to identify the common foodborne pathogens from animal-source foods & by-products with their antimicrobial drug susceptibility and resistance gene profile. The common foodborne pathogens *Escherichia coli* (*E. coli*), *Salmonella*, *Streptococcus*, *Staphylococcus*, and *Campylobacter* species were identified in livestock and poultry food products. The prevalence of foodborne pathogens was found higher in poultry food & by-product compared with livestock (*p* < 0.05). The antimicrobial drug susceptibility results revealed decreased susceptibility to penicillin, ampicillin, amoxicillin, levofloxacin, ciprofloxacin, tetracycline, neomycin, streptomycin, and sulfamethoxazole-trimethoprim whilst gentamicin was found comparatively more sensitive. Regardless of sources, the overall MDR pattern of *E. coli*, *Salmonella*, *Staphylococcus*, and *Streptococcus* were found to be 88.33%, 75%, 95%, and 100%, respectively. The genotypic resistance showed a prevalence of *bla_TEM_*, *bla_SHV_*, *bla_CMY_*, *tetA*, *tetB*, *sul1*, *aadA1*, *aac(3)-IV,* and *ereA* resistance genes. The phenotype and genotype resistance patterns of isolated pathogens from livestock and poultry had harmony and good concordance, and *sul1* & *tetA* resistance genes had a higher prevalence. Good agricultural practices along with proper biosecurity may reduce the rampant use of antimicrobial drugs. In addition, proper handling, processing, storage, and transportation of foods may decline the spread of MDR foodborne pathogens in the food chain.

## 1. Introduction

Foodborne illness is a key public health concern worldwide which occurs due to the ingestion of contaminated food products, mostly animal and poultry-derived food [1,2,3]. The prime cause of foodborne infections is the presence of bacteria in foods which will grow under favorable conditions and produce toxins in food [4]. Currently, foodborne illness caused by bacterial contamination is one of the foremost threats distressing public health [5]. *Staphylococcus aureus*, *Salmonella paratyphoid*, *Campylobacter*, *Listeria monocytogenes* (*L*. *monocytogenes*), and *Escherichia coli* (*E. coli*) are the most important zoonotic bacterial pathogens that cause foodborne illness worldwide, and deaths due to consumption of contaminated animal products [6]. In terms of biological threats, bacterial agents are the utmost severe concern regarding the issues of the supply of pathogen-free animal-derived foods to consumers [7,8]. Animal-derived food products, most commonly red meat, white meat including dairy products, and eggs are the important vehicles through which people may be exposed to foodborne pathogens, especially bacteria [9]. In addition, the poultry vendors at wet markets had poor knowledge of food safety, foodborne pathogens, and zoonoses in Bangladesh is also a key factor for transmitting foodborne pathogens [10]. 

Animal-source foods especially milk, meat, eggs, and their diversified products may become exposed with pathogenic bacteria during harvesting, slaughtering, processing, and marketing [11,12]. Environmental factors such as the nature of pathogens, their hosts (human or animal), host immunity, environmental temperature, etc. have evolved food-derived bacterial pathogens, making the population more susceptible to infection [13]. With rapidly changing human consumption habits, global food markets, and climate change, the fight against bacterial foodborne illness faces new challenges [14]. In this regard, foodborne bacterial infections may be prohibited and controlled by the proper cooking, preparation, and storage of food. In addition, multi-sectoral action is urgently needed to address AMR and to achieve the Sustainable Development Goals (SDGs) [15]. The World Health Organization (WHO) lists AMR as one of the top ten public health threats in the world. The unnecessary use of antibiotics in humans and agriculture has resulted in the widespread growth of antibiotic-resistant strains, and their outbreak in the environment has caused serious health hazards [16,17,18]. The Centers for Disease Control and Prevention (CDC) states that more than 2.8 million people in the United States develop serious infections with antibiotic-resistant bacteria each year, and at least 35,000 die each year from the direct effects of these resistant pathogens [19]. It is evidenced that AMR foodborne pathogens including *E. coli*, *Campylobacter*, *Listeria*, and *Salmonella* are closely linked with chicken meat, beef, and pork [20,21], as well as retail meat [22,23]. 

Extensive use of antibiotics in livestock and poultry production systems is known to contribute to the development of AMR [24]. To date, there is a growing concern about the potential for AMR to be transmitted through the food chain. AMR food pathogens in food-producing animals may infect humans through consumption of contaminated food or water as well as direct contact with animals [25]. In addition, bacteria in foods that are AMR may be more persistent in food processing environments. In this regard, monitoring antimicrobial resistance in the food chain is important for understanding the spread of resistance and making relevant risk assessment data. Previous studies showed the occurrences of foodborne pathogens in poultry meat collected from different sources and selected areas of Bangladesh [26,27,28]. However, no comprehensive research work was carried out on the determination of the antimicrobial drug susceptibility profile of common foodborne pathogens recovered from both livestock and poultry food products and by-products throughout the country. Therefore, considering the paramount importance of antimicrobial resistance globally, the present study was undertaken to isolate and identify foodborne pathogens along with the characterization of their antimicrobial drug susceptibility patterns with resistance genes in animal-derived food & by-product in Bangladesh. The study would help in addressing containment and intervention strategies of MDR foodborne pathogens in the food chain. 

## 2. Results

### 2.1. Prevalence of AMR Pathogens in Animal-Derived Food Products

*E. coli*, *Salmonella*, and *Staphylococcus* species were isolated from both livestock and poultry food product & by-product samples throughout the country. In addition, *Streptococcus* species were isolated from only livestock-source food product & by-product samples. Furthermore, *Campylobacter* species was only isolated from poultry food products of the Chattogram Veterinary and Animal Sciences University (CVASU) component, i.e., from the Chittagong division. The overall prevalence of *E. coli*, *Salmonella*, *Staphylococcus*, and *Streptococcus* in livestock-source food products and by-products were found to be 38.47% (554/1440), 8.26% (119/1440), 14.67% (211/1440), and 4.79% (69/1440), respectively (details of prevalence pattern are presented in Table 1). Similarly, the overall prevalence of *E. coli*, *Salmonella*, *Staphylococcus,* and *Campylobacter* in poultry food products & by-products were 51.59% (454/880), 17.61% (155/880), 21.93% (193/880), and 4.43% (39/880), respectively (Table 2). Regardless of sources, the overall multidrug-resistant pattern of *E. coli*, *Salmonella*, *Staphylococcus*, and *Streptococcus* were found to be 88.33%, 75%, 95%, and 100%, respectively. The prevalence of *E. coli* was found higher in poultry food products & by-products compared with livestock (51.59% vs. 38.47%; *p* < 0.05). Similarly, a higher statistical association was found between the prevalence of *Salmonella* in livestock and poultry food products (17.61% vs. 8.26%; *p* < 0.05). In addition, an association was also found between the prevalence of *Staphylococcus* in livestock and poultry samples (21.93% vs. 14.67%; *p* < 0.05). On the other hand, *Campylobacter* was found in poultry food products & by-products collected from only the Chittagong division and the overall percentage was 4.43% (39/880).

### 2.2. Phenotypic Resistance Pattern

The AST (Antimicrobial Sensitivity Testing) pattern of *E. coli* showed the highest resistance to penicillin (P) (96.19%) followed by ampicillin (AMP) (90.71%), amoxicillin (AMX) (86.87%), oxytetracycline (O) (78.32%), cloxacillin (COX) (70.37%), and sulfamethoxazole-trimethoprim (COT) (70.01%). Among the antimicrobials, gentamicin (GEN) (66.46%) was found to be the most susceptible. The detailed AST pattern of *E. coli* is presented in Figure 1. The AST result of *Salmonella* showed the highest resistance to penicillin (96.15%), followed by ampicillin (AMP) (91.48%), oxytetracycline (O) (82.2%), amoxicillin (73.1%), and cloxacillin (67.85%) whilst the highest susceptibility was recorded to gentamicin (82.91%), followed by ceftriaxone (CTR) (58.88%). The detailed AST pattern of *Salmonella* is presented in Figure 1. On the other hand, the AST pattern of *Staphylococcus* showed the highest resistance to cloxacillin (87.345), followed by amoxicillin (85.39%), penicillin (85.29%), ampicillin (76.28%), streptomycin (S) (79.47%), and oxytetracycline (67.74%) whereas the highest susceptibility was found in gentamicin (66.32%), followed by ciprofloxacin (CIP) (49.46%) and ceftriaxone (49.36%). The detailed AST pattern of *Staphylococcus* is presented in Figure 1. Moreover, the AST pattern of *Streptococcus* indicated greater resistance to penicillin, followed by cloxacillin (86.27%), oxytetracycline (84.5%), streptomycin (78.87%), amoxicillin (73.23%), and sulfamethoxazole-trimethoprim (69.01%) whilst greater susceptibility was observed to ciprofloxacin (70.43%) and gentamicin (42.26%). The detailed AST pattern of *Streptococcus* is shown in Figure 1.

The AST result of *E. coli* showed higher resistance to erythromycin (97.74%), followed by enrofloxacin (EX) (91.63%), oxytetracycline (87.42%), ciprofloxacin (86.46%), sulfamethoxazole-trimethoprim (84.54%), and tetracycline (TE) (75.82%); however, the highest sensitivity was found to gentamicin (49.21%). The detailed AST pattern of *E. coli* is shown in Figure 2. The AST result of *Salmonella* showed higher resistance to ampicillin (93.97%), followed by oxytetracycline (89.04%), tetracycline (87.5%), doxycycline (DO) (85.71%), enrofloxacin (75.86%), erythromycin (E) (72.41%), ciprofloxacin (70%), and sulfamethoxazole-trimethoprim (63.63%); however, the highest sensitivity was found to gentamicin (71.43%) and levofloxacin (LE) (52.81%). The detailed AST pattern of *Salmonella* is outlined in Figure 2. Consecutively, the AST pattern of *Staphylococcus* predicted higher resistance to ampicillin (93.88%), followed by tetracycline (83.61%), erythromycin (81.05%), neomycin (N) (67.91%), sulfamethoxazole-trimethoprim (63.63%), oxytetracycline (57.97%), enrofloxacin (57.97%), and doxycycline (55.56%); however, the highest sensitivity was found to gentamicin (49.02%) and levofloxacin (45.75%). The detailed AST pattern of *Staphylococcus* is presented in Figure 2. Furthermore, the AST pattern of *Campylobacter* species anticipated higher resistance to erythromycin (100%), followed by ciprofloxacin (85%), sulfamethoxazole-trimethoprim (85%), ampicillin (62.5%), and levofloxacin (60%), whilst higher sensitivity was found to doxycycline (72.5%), oxytetracycline (50%), gentamicin (50%), and neomycin (50%). The AST pattern of *Campylobacter* species is presented in Figure 2. 

### 2.3. Genotypic Resistance Pattern of the Isolates Recovered from Both Livestock and Poultry Food Products & By-Products

The genotypic resistance pattern of the isolates showed resistance to ESBL (*bla_TEM_*, *bla_SHV_*, and *bla_CMY_*), tetracycline (*tetA* and *tetB*), sulfonamide (*sul1*), streptomycin (*aadA1*), gentamicin (*aac(3)-IV*), and erythromycin (*ereA*). Similar genotypic trends, with some deviation, were found among the isolated foodborne pathogens of different species. Regardless of sources, all foodborne pathogens possessed antibiotic-resistant genes (*bla_TEM_*, *bla_SHV_*, *bla_CMY_*, *tetA*, *tetB*, *aadA1*, *aac (3)-IV*, and *ereA*) with different percentages. 

The prevalence of antibiotic-resistant genes *sul1*, *tetA*, *tetB*, *aac(3)-IV*, *ereA*, *bla_SHV_*, *bla_CMY_*, *aadA1*, and *bla_TEM_* was found to be 19.7%, 18.1%, 12.1%, 6.3%, 4.9%, 4.1%, 3.3%, 1.7%, and 0.5%, respectively in the *E. coil* isolates. The prevalence of antibiotic-resistant genes *sul1*, *tetA*, *tetB*, *aadA1*, *aac(3)-IV*, *ereA*, *bla_TEM_*, *bla_CMY_*, and *bla_SHV_* was found to be 21.5%, 17.5%, 7%, 3.5%, 3.5%, 3%, 3%, 1.5%, and 0.5%, respectively in the *Streptococcus* isolates. The prevalence of antibiotic-resistant genes *sul1*, *tetA*, *tetB*, *bla_TEM_*, *ereA*, *aadA1*, *aac(3)-IV, bla_CMY_*, and *bla_SHV_* was found to be 39%, 34%, 8.5%, 6.5%, 5.5%, 4.5%, 3.5%, 2.5%, and 1.5%, respectively in the *Salmonella* isolates. The prevalence of antibiotic-resistant genes *sul1*, *tetA*, *mecA*, *tetB*, *bla_SHV_*, *bla_TEM_*, *ereA*, *aadA1*, *aac(3)-IV,* and *bla_CMY_* was found to be 30%, 29.5%, 20%, 5.5%, 4.5%, 3.5%, 3%, 2.5%, 2.5%, and 1.5%, respectively in the *Staphylococcus* isolates. The *sul1* and *tetA* gene was found in higher percentages among the foodborne pathogens compared with other genotypic resistance genes regardless of the sources of collection of pathogens and the type of pathogens (Figure 3).

There was a similar trend in the phenotypic resistance patterns of foodborne pathogens. The phenotypic resistance was found to be comparatively higher in the foodborne pathogens isolated from poultry than livestock-source foods and by-products (*p* < 0.05). The phenotypic and genotypic resistance profiles of various isolates of foodborne pathogens were shown to have a narrower range of variation and variability. The phenotypic and genotypic resistance results indicated that multidrug-resistant and ESBL-producing foodborne pathogens were prevailing in the livestock- and poultry-source food products & by-products in Bangladesh.

## 3. Discussion

Antimicrobials are frequently used to treat infectious diseases in both humans and animals [29]. Recently, the overuse and misuse of antibiotics in livestock have become a great concern for public health authorities. Contrarily, because withdrawal periods before harvesting or marketing livestock products have been ignored, antibiotic residues are now another rising concern to public health [30]. The main consequence of antibiotic residues in animal-derived foods is the enhancement of the development of antimicrobial resistance. The presence of antibiotic-resistant foodborne pathogens in food may lead to gastrointestinal disorders in human beings [31]. On the other hand, antibiotic-resistant pathogens may transfer the gene to other microorganisms through vertical and horizontal transmission [29,32] resulting in the spread of AMR pathogens. Several previous studies have shown the emergence of multi-resistant bacterial pathogens from a wide variety of sources in the food chain, increasing the need for proper use of antibiotics in both the veterinary and human health sectors [33,34,35]. MDR pathogens may cause difficult-to-treat illnesses, increased mortality, and financial burden. Furthermore, infections caused by MDR pathogens are considered a major global public health crisis by the World Health Organization, as the discovery of effective antibiotics has not kept pace with the increase in bacterial antibiotic resistance [36]. The demand for high-value animal products such as milk, meat, and eggs has increased due to economic solvency, rapid urbanization, and changing food habits of nations [37]. Foodborne pathogens can enter into the food cycle during production, processing, and marketing. Humans can get antibiotic-resistant bacterial infections in many ways, including ingestion of contaminated food or contact with colonized or diseased animals, body fluids, excretions, or secretions [38]. In addition, the pathogens can cause illness due to the consumption of undercooked food and produce illnesses either by their presence or by-production of toxins, or both. 

The important foodborne pathogens of animal-source foods that have been globally identified are *Salmonella*, *Campylobacter*, *E. coli*, and *Staphylococcus* [20,22], and these trends are apparent in the current study in a similar fashion. Per the previous reports from home and abroad [39,40,41,42,43,44,45], the AST results of *E. coli* isolated from livestock and poultry showed a wider range of resistance to penicillin (100%), tetracycline (72–100%), oxytetracycline (78–93%), sulfamethoxazole-trimethoprim (51–88%), ampicillin (89.5–100%), amoxicillin (92–95%), streptomycin (19–70%), erythromycin (89%), ciprofloxacin (50%), chloramphenicol (43–50%), gentamicin (8–28%), enrofloxacin (55%), and norfloxacin (50%). In contrast, the phenotypic resistance pattern of *E. coli* to various antimicrobial agents recovered from both livestock and poultry in the present study also showed a similar trend where the AST pattern of *E. coli* showed the highest resistance to penicillin, followed by ampicillin, amoxicillin, oxytetracycline, cloxacillin, and sulfamethoxazole-trimethoprim. Among the antibiotics, gentamicin possessed the lowest resistance percentage, which is also comparable to the previous studies stated above. *E. coli* is one of the major pathogenic microorganisms that may reach animal-derived foods and is an indication of contamination by manure, soil, and contaminated water [46]. *E. coli* are commensal bacteria, and *E. coli* pathotypes can cause zoonotic disease that poses a public health risk [47]. In addition, Shiga toxin-producing *E. coli* is associated with the development of several life-threatening infections in humans [48]. In this regard, our recently published data showed that the most common class of antimicrobials used in large animal farms were Penicillin (61.79%), Oxytetracycline (55.66%), Sulfa drug (55.66%), Streptomycin (54.72%), followed by Ciprofloxacin (51.89%), Gentamicin (43.13%), and Ceftriaxone (34.91%) [30]. These data indicate that the bacteria became more resistant to such most commonly used antimicrobials in the study area.

Similarly, the AST result of *Salmonella* recovered from poultry revealed a wider range of resistance to ciprofloxacin (70–88%), ampicillin (66–75%), tetracycline (77–84%), gentamicin (33–68%), nalidixic acid (22–60%), and streptomycin (44–77%) [49]. While lower resistance (5–8%) was observed to chloramphenicol, azithromycin, imipenem, amikacin, and sulfamethoxazole-trimethoprim [49]. In contrast, in our present study, the AST result of *Salmonella* showed the highest resistance to penicillin, followed by ampicillin, oxytetracylcine, amoxicillin, and cloxacillin whilst the highest susceptibility was recorded to gentamicin, followed by ceftriaxone. *Salmonella* is widespread in nature [50] and is the most important pathogenic bacterium in both humans and animals [51]. Bacterial pathogens are more commonly found during outbreaks of foodborne disease [52] and are accountable for around 93.8 million foodborne illnesses and 155,000 fatalities annually worldwide [53]. 

According to the previous study reports, the AST result of *Streptococcus* in livestock and poultry showed a wider degree of resistance to streptomycin (70–100%), amoxicillin (30–100%), and ampicillin (60–100%) [39]. Sequentially, the AST result of *Staphylococcus* revealed a broader range of resistance to penicillin (82–100%), amoxicillin (42–100%), ampicillin (97%), streptomycin (70–100%), oxytetracycline (74–78%), ciprofloxacin (17–50%), sulfamethoxazole-trimethoprim (30%), gentamicin (18%) cefixime (73.9%), cloxacillin (82.6%), and oxacillin (56–98%) [44,54,55]. In contrast, the present study results from the AST pattern of *Streptococcus* indicated a greater resistance to penicillin, followed by cloxacillin, oxytetracycline, streptomycin, amoxicillin, and sulfamethoxazole-trimethoprim whilst greater susceptibility was observed to ciprofloxacin and gentamicin. The data from the present study suggested that gentamicin was found to be susceptible to isolated common foodborne pathogens.

*Staphylococcus* are commensal bacteria that are normal inhabitants of the skin, nose, and mucous membranes of healthy people and animals [56,57]. However, it is also known as an opportunistic foodborne pathogen [58] that may cause several infectious diseases with different degrees of severity [56]. It causes numerous infections in humans and animals [59]. The presence of *S. aureus* in food products is an alarming and serious threat to public health in terms of food safety when it releases toxins and causes illness [60]. Methicillin-resistant *S. aureus* (MRSA) has emerged due to the unnecessary use of antibiotics [61,62]. The presence of MRSA in farm animals and the potential for cross-contamination in humans have been a great concern [63].

*Campylobacter* are the leading cause of foodborne diarrhea in humans worldwide [64], which is mainly due to contamination of food of animal origin [65]. *Campylobacter* can colonize in warm-blooded animals and poultry [66]. The zoonotic nature of *Campylobacter* species makes it clinically and economically important worldwide [67]. It has accounted for 15% of food-related illnesses leading to hospital admissions and 6% resulting in death; about 400 million cases are reported each year due to foodborne infection [68,69]. The economic impact of *Campylobacter* infections is mainly related to the treatment cost, production loss, and pathogen control expenses [67]. 

The ESBL-producing foodborne pathogens were previously identified by researchers from a variety of sources of livestock and poultry [70,71,72]. More recent studies have shown that ESBLs-producing bacteria frequently colonized in poultry [73,74,75] and cattle [76,77]. On the other hand, tetracycline-resistant genes are commonly encoded by plasmids & transposons and are transmitted by conjugation. However, in some isolates, the corresponding gene is also found on the chromosome [78,79]. Mechanisms of resistance to tetracycline by the acquisition of the *tet* gene primarily include efflux pumping, ribosome protection, and enzymatic inactivation. The resistance of gram-negative bacteria to sulfonamides is associated with the presence of the *sul* gene, which encodes dihydropteroate synthase in a manner that the drug cannot inhibit. The *sul* gene has already been identified in *Enterobacteriaceae*, especially in the genera *Escherichia* and *Salmonella* [80]. In this regard, the present study finding showed that the *sul1* and *tetA* genes were found in higher percentages among the foodborne pathogens compared with other resistance genes regardless of the sources of isolation of the pathogen and the type of pathogens. 

To the best of our knowledge, for the first time, this study was conducted throughout the country and found that multidrug-resistant and ESBL-producing foodborne pathogens were prevailing in the livestock- and poultry-source food products & by-products in Bangladesh. However, the present study has some limitations. In this study, we did not collect the environmental samples which may be contaminated by the livestock- and poultry-source food products and by-products. In addition, the sampling area was limited in each division. Further detailed studies with larger samples size from each district of Bangladesh are needed. Details of further phenotypic & genotypic analysis in a wider range with 16S rRNA sequence profiling of these isolates would help the scientists in this field to combat AMR as well as to stop the spreading of MDR foodborne pathogens to humans.

## 4. Materials and Methods

### 4.1. Study Area

The study was conducted in thirty-three districts under the eight administrative divisions of Bangladesh. The seven components (educational institutes) of the project covered each division with at least four districts while the component (educational institute) Patuakhali Science and Technology University covered two divisions with nine districts. The seven components (educational institutes) of the project are the Bangladesh Agricultural University (BAU), Bangladesh Livestock Research Institute (BLRI), Rajshahi University (RU), Patuakhali Science and Technology University (PSTU), Chattogram Veterinary and Animal Sciences University (CVASU), Sylhet Agricultural University (SAU), and Haji Danesh Science and Technology University (HSTU). Figure 4 shows the study divisions followed by districts area covered by the seven components (educational institutes). The study protocol was authorized by the Animal Welfare and Experimentation Ethics Committee of the Bangladesh Agricultural University, Mymensingh, (approval number: AWEEC/BAU/2018(17)).

### 4.2. Sampling Design

A total of 2320 samples were collected across the country, of which 880 were taken from poultry and 1440 from large & small ruminants in thirty-three districts of eight divisions of Bangladesh by the seven components of the project. Poultry source samples were broiler meat (*n* = 160), layer meat (*n* = 160), egg (*n* = 240), broiler feces (*n* = 160), and layer feces (*n* = 160). All poultry sources samples were directly collected from layer and broiler farms. On the other hand, large & small ruminant samples were cattle meat (*n* = 160), sheep or goat meat (*n* = 160), buffalo meat (*n* = 80), cattle raw milk (*n* = 200), sheep or goat raw milk (*n* = 200), buffalo raw milk (*n* = 120), cattle feces (*n* = 200), sheep or goat feces (*n* = 200), and buffalo feces (*n* = 80). Different types of meat samples were collected for local raw meat markets; however, raw milk and feces from different animals were collected from farms. All the samples were collected in aseptic condition using sterile instruments and carefully transferred into sterile Eppendorf tubes (for raw milk) or zipper bags (for solid samples) from animal and poultry farms. Immediately after collection, samples were kept in a transport box for maintaining a 4 °C temperature. The samples were then transported to the bacteriological laboratory of the Department of Microbiology and Hygiene, Faculty of Veterinary Science, BAU, Mymensingh for microbiological analysis. According to our previous study [30], the Raosoft sample volume calculation method was used to determine the sample size with a 5% margin of error and 95% confidence level. 

### 4.3. Conventional Culture Method

*E. coli*, *Salmonella*, *Streptococcus,* and *Staphylococcus* were targeted for the isolation from livestock whilst *E. coli*, *Salmonella*, *Staphylococcus,* and *Campylobacter* were targeted for the isolation from poultry sources following the standard procedure as applied earlier [44,45,55,81,82,83,84,85]. Briefly, 0.5 g of each sample was inoculated in nutrient broth and incubated at 37 °C for 12 h for the initial growth of *Escherichia coli*, *Salmonella,* and *Staphylococcus aureus*. The cultures from nutrient broth were streaked on Eosin methylene blue (EMB) agar (Hi media, Maharashtra, India), Salmonella-Shigella (SS) agar (Hi media, Maharashtra, India), and Mannitol salt agar (MSA) (Hi media, India) plates using platinum loop for the isolation of *E. coli*, *Salmonella*, and *S. aureus*, respectively. Milk samples (200 µL) were also inoculated in Kenner Fecal (KF) Streptococcal broth (Hi media, India) for the initial growth of *Streptococcus*, then streaked on KF Streptococcal agar media for the isolation of *Streptococcus*. Then all the culture plates were incubated at 37 °C for 24 h. For the isolation of *Campylobacter*, all samples were directly inoculated on selective campylobacter base agar (Oxoid Ltd., Hampshire RG24 8PW, UK) containing antibiotics (Amphotericin B has been added to suppress the growth of yeast and fungal contaminants that may occur at 37 °C, and improved selectivity was achieved by adding cefoperazone) and 5–7% sheep blood [86]. The plates were incubated in an anaerobic jar (Oxoid™ AnaeroJar™ 2.5 L) under microaerophilic conditions with a CO_2_ sachet (Thermo Scientific TM Oxoid Anaero Gen 2.5 L sachet) (10% CO_2_, 95% humidity) at 42 °C for three days. After 72 h, single characteristic (small, round, creamy-gray, or whitish) colonies from each plate were selected and inoculated in tryptic soy broth (Oxoid Ltd., Hampshire RG24 8PW, UK) and incubated at 37 °C for three days under microaerophilic conditions. The single colonies of suspected bacteria were again inoculated in selective broth and streaked on selective agar media to obtain a pure culture. 

### 4.4. Molecular Detection

Whole genomic DNA was extracted from each pure culture by a conventional boiling method [49,85]. Following the boiling method, the DNA was measured using spectrophotometers. PCR was performed for confirmatory detection of each isolate using specific primers (Table 3) and the PCR condition for each bacterial species was used following the standard operating procedure described by different authors (Table 3).

### 4.5. Determination of Phenotypic Resistance Pattern

Antimicrobial susceptibility testing was performed using the Kirby Bauer disk diffusion method in adherence with the guidelines of the Clinical and Laboratory Standards Institute [91]. Briefly, fresh colonies were suspended in saline and the turbidity of the suspension was measured in comparison with the 0.5 McFarland standards (approximately 1.5 × 10^6^ CFU/mL). The bacterial suspension was smeared on the surface of Mueller-Hinton (MH) agar (Oxoid Ltd., Hampshire RG24 8PW, UK) and an antibacterial disc with a disc dispenser was placed on it within 15 min and the plate was incubated at 37 °C for 24 h. The zone of inhibition adjacent to the disks was measured and compared with the breakpoints of CLSI. A number of 16 different antimicrobial disks (Hi media, India) were used for AST of all four foodborne pathogens such as penicillin (P, 10 units), ampicillin (AMP, 10 μg), amoxicillin (AMX, 30 μg), cloxacillin (COX, 5 μg), ceftriaxone (CTR, 30 μg), tetracycline (TE, 30 μg), doxycycline (DO, 30 μg), oxytetracycline (O, 30 μg), sulfamethoxazole-trimethoprim (COT, 25 μg), gentamicin (GEN, 10 μg), erythromycin (E, 15 μg), ciprofloxacin (CIP, 5 μg), streptomycin (S, 10 μg), levofloxacin (LE, 5 μg), enrofloxacin (EX, 5 μg), and neomycin (N, 30 µg). Based on their common therapeutic usage at the field level in the study areas, different antimicrobial disks for livestock and poultry sources bacteria were chosen [30]. The criteria developed by Magiorakos et al. [92] were used to define multidrug-resistant (MDR) bacteria. Susceptible, intermediate, and resistant were defined according to the new DIN EN ISO 20776-1 standard [93], which is valid worldwide. The sensitivity of a bacterial strain to a given antibiotic is said to be intermediate when it is inhibited in vitro by a concentration of this drug that is associated with an uncertain therapeutic effect [94]. The *Escherichia coli* ATCC 25922 strain served as a validated positive control.

### 4.6. Determination of Genotypic Resistance Pattern

*E. coli*, *Salmonella*, *Staphylococcus*, *Streptococcus*, and *Campylobacter* isolates were screened by PCR for the detection of Extended Spectrum β-Lactamases (ESBL) genes (*bla_TEM_*, *bla_SHV_*, and *bla_CMY_*), tetracycline-resistant genes (*tet*A and *tet*B), sulfonamide-resistant gene (*sul1*), streptomycin-resistant gene (*aadA1*), gentamicin-resistant gene (*aac(3)-II*), neomycin-resistant gene (*aph(3)-I*), and erythromycin-resistant gene (*ereA*). The list of primers to detect resistance genes is given in Table 4. 

## 5. Conclusions

The phenotypic and genotypic resistance profiles uncovered by the present study indicated that MDR- and ESBL-producing foodborne pathogens were prevailing in the livestock- and poultry-source food products & by-products in Bangladesh. MDR foodborne pathogens are a current public health concern worldwide including in Bangladesh. Foodborne pathogens are usually spread due to improper handling, processing, preparation, and storage of food. The unnecessary use of antimicrobials in farm practices is the main driver for the emergence of antimicrobial-resistant pathogens in the livestock and poultry value chain. Good agricultural practices, good veterinary practices, good manufacturing practices, and proper farm biosecurity are important tools to curb the development of AMR pathogens in livestock and poultry food products. Moreover, policy intervention, stakeholder awareness, motivation, training, advocacy, and mass media dissemination are the imperative pathways to combat the spread of foodborne pathogens and AMR. 

## Figures and Tables

**Figure 1 antibiotics-11-01551-f001:**
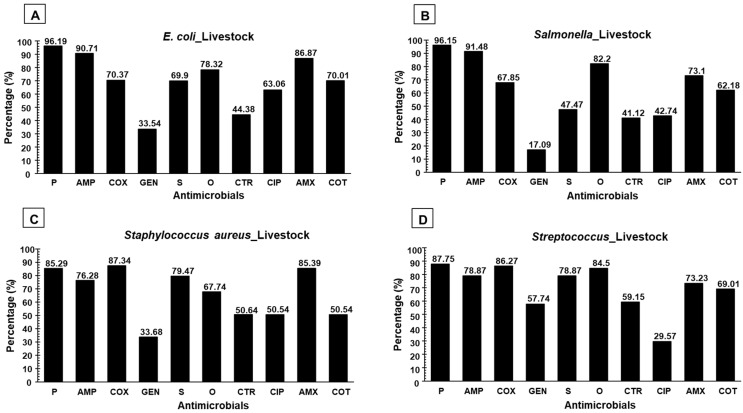
Phenotypic resistance pattern to various antimicrobial agents, AST pattern of (**A**) *E. coli*, (**B**) *Salmonella*, (**C**) *Staphylococcus*, and (**D**) *Streptococcus* recovered from livestock food products & by-products.

**Figure 2 antibiotics-11-01551-f002:**
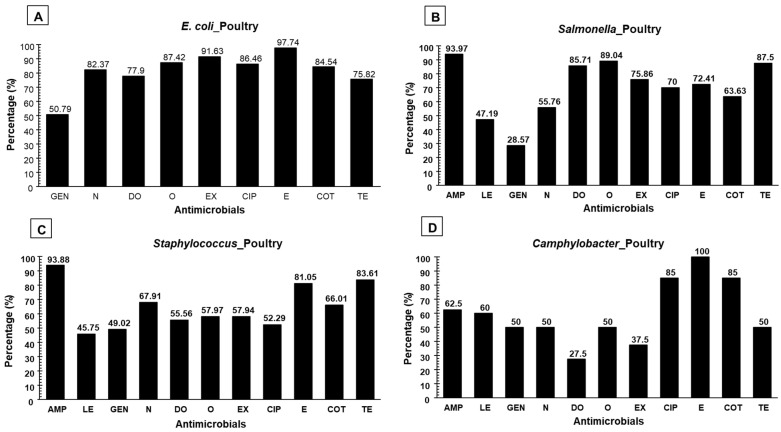
Phenotypic resistance pattern to various antimicrobial agents, AST pattern of (**A**) *E. coli*, (**B**) *Salmonella*, (**C**) *Staphylococcus*, and (**D**) *Campylobacter* recovered from poultry food products & by-products.

**Figure 3 antibiotics-11-01551-f003:**
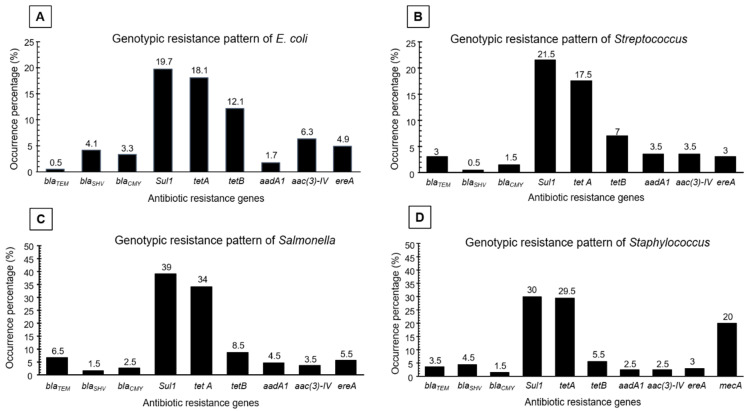
Genotypic resistance pattern of (**A**) *E. coli*, (**B**) *Streptococcus*, (**C**) *Salmonella*, and (**D**) *Staphylococcus* recovered from both livestock and poultry food products & by-products.

**Figure 4 antibiotics-11-01551-f004:**
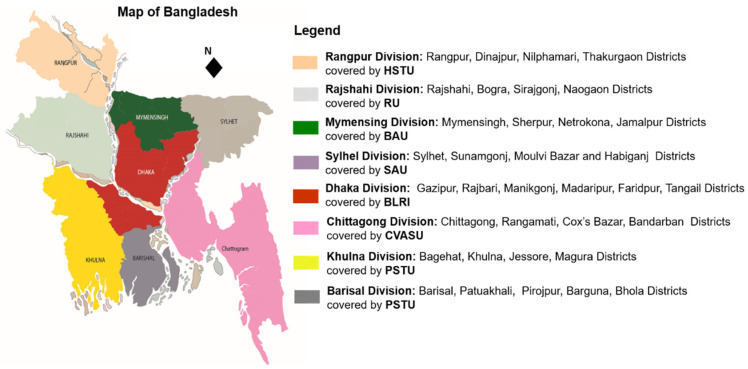
AMR surveillance study areas covered by seven components (educational institutes).

**Table 1 antibiotics-11-01551-t001:** Prevalence of AMR pathogens in livestock-source food products and by-products.

Organisms	Positive Samples (Total Samples)	Prevalence (%)	Confidence Interval (95% Cl)
*E. coli*	554 (1440)	38.47	35.99–41.01
*Salmonella*	119 (1440)	8.26	6.95–9.80
*Staphylococcus*	211 (1440)	14.65	12.92–16.57
*Streptococcus*	69 (1440)	4.79	3.80–6.02

**Table 2 antibiotics-11-01551-t002:** Prevalence of AMR pathogen in poultry source food products and by-products.

Organisms	Positive Samples (Total Samples)	Prevalence (%)	Confidence Interval (95% Cl)
*E. coli*	454 (880)	51.59	48.29–54.88
*Salmonella*	155 (880)	17.61	15.24–20.27
*Staphylococcus*	193 (880)	21.93	19.32–24.78
*Campylobacter*	39 (880)	4.43	3.26–6

**Table 3 antibiotics-11-01551-t003:** List of primers used for bacterial species identification.

AMR Pathogens	Primers	Sequence (5′–3′)	Amplicon Size (bp)	Reference
*Salmonella*	F	TCATCGCACCGTCAAAGGAACC	284	[87]
	R	GTGAAATTATCGCCACGTTCGGGCAA		
*E. coli*	F	CCCCCTGGACGAAGACTGAC	401	[88]
	R	ACCGCTGGCAACAAAGGATA		
*Staphylococcus*	F	CCTGAAACAAAGCATCCTAAAAA	155	[89]
	R	TAAATATACGCTAAGCCACGTCCAT		
*Campylobacter*	F	ATCTAATGGCTTAACCATTAAAC	857	[90]
	R	GGACGGTAACTAGTTTAGTATT		
*Streptococcus*	F	AGCGGGGGATAACTATTGGA	569	[84]
	R	TACGCATTTCACCGCTACAC		

**Table 4 antibiotics-11-01551-t004:** List of primers used to detect resistant genes.

Class	Target Gene	Primers	Sequence (5′–3′)	Amplicon Size (bp)	Reference
Gentamicin	*aac(3)-IV*	F	CTTCAGGATGGCAAGTTGGT	286	[95]
		R	TCATCTCGTTCTCCGCTCAT		
Tetracycline	*tetA*	F	GGTTCACTCGAACGACGTCA	577	[95]
		R	CTGTCCGACAAGTTGCATGA		
	*tetB*	F	CCTCAGCTTCTCAACGCGTG	634	
		R	GCACCTTGCTGATGACTCTT		
Beta lactams	*bla_TEM_*	F	ATAAAATTCTTGAAGAC	1073	[96]
		R	TTACCAATGCTTAATCA		
Beta lactams	*bla_SHV_*	F	TCGCCTGTGTATTATCTCCC	768	[95]
		R	CGCAGATAAATCACCACAATG		
Beta lactams	*bla_CMY_*	F	TGGCCAGAACTGACAGGCAAA	462	[95]
		R	TTTCTCCTGAACGTGGCTGGC		
Erythromycin	*ereA*	F	GCCGGTGCTCATGAACTTGAG	419	[95]
		R	CGACTCTATTCGATCAGAGGC		
Sulfonamide	*sul1*	F	TTCGGCATTCTGAATCTCAC	822	[95]
		R	ATGATCTAACCCTCGGTCTC		
Streptomycin	*aadA1*	F	TATCCAGCTAAGCGCGAACT	447	[95]
		R	ATTTGCCGACTACCTTGGTC		
Methicillin-resistant	*mecA*	F	AAAATCGATGGTAAAGGTTGGC	533	[97]
		R	AGTTCTGGAGTACCGGATTTGC		

## Data Availability

Not applicable.

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
