# Peer review of "Antimicrobial Resistance Profile of Common Foodborne Pathogens Recovered from Livestock and Poultry in Bangladesh"

_antibiotics, 2022, doi:10.3390/antibiotics11111551_

Round 1
Reviewer 1 Report
The authors presenting "Antimicrobial Resistance Profile of Common Foodborne Pathogens Recovered from Livestock and Poultry in Bangladesh" They mostly focused on AST in livestock and livestock products from different locations in Bangladesh. However, the study is limited only to few bacteria and few ARGs. The information is piled up in the manuscript in a poorly organized manner. There is no line number in the manuscript which is very difficult to review. The authors missed the prevalence of AMR to common and important drugs like Carbapenem, Colistin, and tigecylcine which are considered the last therapeutic drugs for MDR bacteria. While in ESBL the authors also missed blaCTX-M gene which is the most commonlyoccuring B-lactamase. The manuscript need extensive editing. The discussion part is very poor. The first 3 paragraphs in discussion is just information about AMR which should be merge into one and write only the important materials.
Some of the major and minor comments I mentioned are below. I will also attached the manuscript I highlighted which will be easy for revision.
Introduction:
1. Page 2. Pargaraph No 1: “Foodborne intoxication occurs due to consumption of various ……………………………………. and will produce toxins in food” These two sentences are repetition, should be combine into one and rephrase it.
2. Page 2. Pargaraph No 1: “In terms of biological threats, bacterial agents are the utmost severe disquiet concerning the issues of the supply of pathogen free meat to the consumers” ….. only meat? or milk and other products. Not clear
3. Page 2. Pargaraph No 1: “Vertebrate species are a natural reservoir of many pathogens that may be transmitted through food and may cause infections in humans” No need for this sentence, suggested to remove.
4. Page 2. Pargaraph No 2: “Animal origin foods especially milk, meat, eggs and their diversified products may become expose with pathogenic bacteria during harvesting, slaughtering, processing and marketing” Add the following relevant reference.
https://dx.doi.org/10.17582/journal.pjz/20190513220514
5. Page 2. Pargaraph No 2: “Environmental factors have evolved food-derived bacterial pathogens, making the population more susceptible to infection” which environmental factors ? please elaborate
6. Page 2. Pargaraph No 2: Sustainable Development Goals (SDGs). Reference is missing.
7. Page 2. Pargaraph No 2: “It is evidenced that AMR foodborne pathogens including E. coli, Campylobacter, Listeria, and Salmonella closely linked with chicken meat, beef and pork [18] as well as retail meat [19] have been reported.” Add the relevant reference here mentioned below;
DOI: 10.1016/j.micpath.2020.104722
8. Page 2. Pargaraph No 3: “Antibiotic misuse and or overuse in livestock and poultry production systems are known to contribute to the development of AMR [20].” Please rephrase the sentence.
Results
9. Page 3. Section: “2.1. Prevalence of AMR pathogen in animal derived food products” pathogen or pathogens?
10. Page 3. Section: “2.1 “E. coli, Salmonella, Staphylococcus, and Streptococcus species were isolated from live[1]stock origin food & by product samples through the country. On the other hand, E. coli, Salmonella, and Staphylococcus species were isolated from poultry origin food samples through the country” Merge these two sentences into one.
11. Page 3. Section: “2.1 “The overall prevalence of E. coli, Salmo[1]nella, Staphylococcus, and Streptococcus in livestock origin food products and by-products were found to be 38.47% (554/1440), 8.26% (119/1440), 14.67% (211/1440) and 4.79% (69/1440), respectively.” It should be specific that which bacteria was isolated from which animal or biproduct.
12. Table 1: The authors need to mention the prevalence of each bacteria not just in livestock also mention which animal, which biproduct.........
13. Table 2: Need complete information not just poultry and livestock origin. As you are focusing on food products. Need clarity in the results
14. Page 3. Section 2.2. “Phenotypic resistance pattern” How you checked phenotypic resistance? What criteria you have set that you found lower susceptibilitiy to these mentioned drugs?
15. Page 3. Section 2.2.1. “AST (Antimicrobial Sensitivity Testing)” No need for extra section. You should combine these two parts as both are about phenotypic resistance patterns and delete non-essential results.
Discussion:
16. Page 6. Paragraph 1: “Antimicrobials are generally used for the therapeutic purposes of infectious diseases of human and animals” Rephrase the sentence and the whole manuscript need extensive proof editing.
17. Page 6. Paragraph 1: “On the other hand, antibiotic residues are becoming another emerging threat of public health due to ignoring of withdrawal periods prior harvesting or marketing of livestock products” This sentence also need to rephrase
18. Page 6. Paragraph 1: “The main consequences of antibiotic residues in animal derived-foods is the enhancing of the development of resistant pathogen.” Are you sure? How the drug residues enhancing resistance in pathogens.
19. Page 6. Paragraph 2: This paragraph should be merge with the first para of discussion and write according to the context. No need to write common things.
20. Page 7. Paragraph 3: I don’t understand why the authors just repeatedly discussing other studies in the discussion part. In the first 3 paragraphs the authors only discussed about the importance of AMR and details of food borne pathogens which i think is un necessary. You need to write the first paragraph about the importance of this research. You need to merge the first 3 para into one and delete unnecessary and repeated things. Just write something specific. And from second para you need to start from your results and compare it with other similar studies.
21. Page 7. Paragraph 4: I did not understand that authors are presenting here the results of this study or other studies? Please write it clearly.
22. Page 7 Paragraph 5: “Similarly, the AST result of Salmonella recovered from poultry revealed wider range of resistance to ciprofloxacin (70-88%), …………………… 93.8 million foodborne illnesses and 155,000 fatalities annually worldwide [46].” The authors only compared the AST results of other with this study but did not explained why there is a difference. The discussion part need extensive revision otherwise it is not suitable to publish.
Materials and Methods
23. Page 9. Section 4.2. Sampling design: Please explain the samples details. How and from where the meat and milk samples were collected? From local market or from slaughterhouses?
24. Page 9. Section 4.3. 4.3. Conventional culture method: “Briefly, each of the samples was inoculated in nutrient broth and incu[1]bated at 37°C for 12 hours for initial growth of Escherichia coli, Salmonella and Staphylococ[1]cus aureus. Briefly, 0.5 gram of each sample was inoculated in nutrient broth and incubated at 37°C for 12 hours for initial growth of Escherichia coli, Salmonella and Staphylococcus…..” These two sentences are repeated twice
25. Page 10. Section 4.3. 4.3. Conventional culture method: “For the isolation of Campylobac[1]ter, all samples were directly inoculated on selective campylobacter base agar (Oxoid Ltd, UK) containing antibiotics and 5-7% sheep blood.” Which antibiotics ??????
26. Page 10. Section 4.4. Molecular detection: Better to change chromosomal DNA into "Whole genomic DNA"
27. Page 10. Section 4.4. Molecular detection: “According to our previous studies [42], we found boiling method is effec[1]tive well-established method for chromosomal DNA extraction as teratogenic technique instead of using commercial kit.” Delete this sentence
28. Page 10. Section 4.5. Determination of phenotypic resistance pattern” “The zone of inhibition adjacent to the disks was measured using conventional methods and compared to the break points of CLSI.” which conventional methods?
29. Page 11. Section 4.5. Determination of phenotypic resistance pattern: “Based on our filed survey [23],” Field or Filed ????????
30. Page 11. Section 4.5. Determination of phenotypic resistance pattern: “Multidrug resistance (MDR) bacteria are defined using the criteria as established by Magiorakos et al. [84]” Rephrase this sentence
31. Page 11. Section 4.5. Determination of phenotypic resistance pattern: “The Escherichia coli ATCC 25922 strain was used as known positive control.” Rephrase this sentence
32. Page 11. Section 4.6. Determination of genotypic resistance pattern: Why the authors detect ARGs of few drugs only? Why not they include Carbapenem, Tigecycline, and Colistin related genes? This is the major gap in your study that you missed the important drugs prevalence which is very common in animals and humans and all these drugs are the last resorts for AMR.
33. Page 11. Section 4.6. Determination of genotypic resistance pattern: “Extended Spectrum β-Lactamases (ESBL) genes (blaTEM, blaSHV, and blaCMY)” Where is blaCTX-M which is the main ESBL gene? Why you only choose just blaTEM, SHV and CMY? Again here you missed the most important and common ESBL gene blaCTX-M.
34. Table 4: Would be better to move this table to supplementary as well the references of this table.

Author Response
Reviewer#1
The authors presenting "Antimicrobial Resistance Profile of Common Foodborne Pathogens Recovered from Livestock and Poultry in Bangladesh" They mostly focused on AST in livestock and livestock products from different locations in Bangladesh. However, the study is limited only to few bacteria and few ARGs. The information is piled up in the manuscript in a poorly organized manner. There is no line number in the manuscript which is very difficult to review. The authors missed the prevalence of AMR to common and important drugs like Carbapenem, Colistin, and tigecylcine which are considered the last therapeutic drugs for MDR bacteria. While in ESBL the authors also missed blaCTX-M gene which is the most commonlyoccuring B-lactamase. The manuscript need extensive editing. The discussion part is very poor. The first 3 paragraphs in discussion is just information about AMR which should be merge into one and write only the important materials.
Response: We appreciate your overall comments. Based on you comments and suggestions we extensively revised the entire manuscript point by point. And are shown in tract change mode.
Some of the major and minor comments I mentioned are below. I will also attached the manuscript I highlighted which will be easy for revision.
Introduction:
- Page 2. Pargaraph No 1: “Foodborne intoxication occurs due to consumption of various ……………………………………. and will produce toxins in food” These two sentences are repetition, should be combine into one and rephrase it.
Response: we did it
- Page 2. Pargaraph No 1: “In terms of biological threats, bacterial agents are the utmost severe disquiet concerning the issues of the supply of pathogen free meat to the consumers” ….. only meat? or milk and other products. Not clear
Response: revised the sentences and make it clear for the reader.
- Page 2. Pargaraph No 1: “Vertebrate species are a natural reservoir of many pathogens that may be transmitted through food and may cause infections in humans” No need for this sentence, suggested to remove.
Response: we remove it
- Page 2. Pargaraph No 2: “Animal origin foods especially milk, meat, eggs and their diversified products may become expose with pathogenic bacteria during harvesting, slaughtering, processing and marketing” Add the following relevant reference.
https://dx.doi.org/10.17582/journal.pjz/20190513220514
Response: Reference added
- Page 2. Pargaraph No 2: “Environmental factors have evolved food-derived bacterial pathogens, making the population more susceptible to infection” which environmental factors ? please elaborate
Response: we included the environmental factors with reference.
- Page 2. Pargaraph No 2: Sustainable Development Goals (SDGs). Reference is missing.
Response: Reference added
- Page 2. Pargaraph No 2: “It is evidenced that AMR foodborne pathogens including E. coli, Campylobacter, Listeria, and Salmonella closely linked with chicken meat, beef and pork [18] as well as retail meat [19] have been reported.” Add the relevant reference here mentioned below;
DOI: 10.1016/j.micpath.2020.104722
Response: Two References added
- Page 2. Pargaraph No 3: “Antibiotic misuse and or overuse in livestock and poultry production systems are known to contribute to the development of AMR [20].” Please rephrase the sentence.
Response: we revised the sentence
Results
- Page 3. Section: “2.1. Prevalence of AMR pathogen in animal derived food products” pathogen or pathogens?
Response: Sorry for our careless mistake and was corrected
- Page 3. Section: “2.1 “E. coli, Salmonella, Staphylococcus, and Streptococcus species were isolated from live[1]stock origin food & by product samples through the country. On the other hand, E. coli, Salmonella, and Staphylococcus species were isolated from poultry origin food samples through the country” Merge these two sentences into one.
Response: we did it
- Page 3. Section: “2.1 “The overall prevalence of E. coli, Salmo[1]nella, Staphylococcus, and Streptococcus in livestock origin food products and by-products were found to be 38.47% (554/1440), 8.26% (119/1440), 14.67% (211/1440) and 4.79% (69/1440), respectively.” It should be specific that which bacteria was isolated from which animal or biproduct.
Response: Basically, we have so many data and have the plan to submit as another article in future, therefore we presented the data as we did in our present manuscript
- Table 1: The authors need to mention the prevalence of each bacteria not just in livestock also mention which animal, which biproduct.........
Response: we did it
- Table 2: Need complete information not just poultry and livestock origin. As you are focusing on food products. Need clarity in the results
Response: clarified it
- Page 3. Section 2.2. “Phenotypic resistance pattern” How you checked phenotypic resistance? What criteria you have set that you found lower susceptibilitiy to these mentioned drugs?
Response: we revised the entire paragraph and deleted some sentences
- Page 3. Section 2.2.1. “AST (Antimicrobial Sensitivity Testing)” No need for extra section. You should combine these two parts as both are about phenotypic resistance patterns and delete non-essential results.
Response: we revised the entire paragraph and deleted some sentences
Discussion:
- Page 6. Paragraph 1: “Antimicrobials are generally used for the therapeutic purposes of infectious diseases of human and animals” Rephrase the sentence and the whole manuscript need extensive proof editing.
Response: we revised the sentence.
- Page 6. Paragraph 1: “On the other hand, antibiotic residues are becoming another emerging threat of public health due to ignoring of withdrawal periods prior harvesting or marketing of livestock products” This sentence also need to rephrase
Response: we revised the sentence
- Page 6. Paragraph 1: “The main consequences of antibiotic residues in animal derived-foods is the enhancing of the development of resistant pathogen.” Are you sure? How the drug residues enhancing resistance in pathogens.
Response: we revised the sentences and the residues in low doses fight against bacteria and is responsible to development of resistant bacteria.
- Page 6. Paragraph 2: This paragraph should be merge with the first para of discussion and write according to the context. No need to write common things.
Response: we revised the entire 3 para and merge as one para.
- Page 7. Paragraph 3: I don’t understand why the authors just repeatedly discussing other studies in the discussion part. In the first 3 paragraphs the authors only discussed about the importance of AMR and details of food borne pathogens which i think is un necessary. You need to write the first paragraph about the importance of this research. You need to merge the first 3 para into one and delete unnecessary and repeated things. Just write something specific. And from second para you need to start from your results and compare it with other similar studies.
Response: we revised the entire 3 para and merge as one para.
- Page 7. Paragraph 4: I did not understand that authors are presenting here the results of this study or other studies? Please write it clearly.
Response: we revised the sentences and make it clear.
- Page 7 Paragraph 5: “Similarly, the AST result of Salmonella recovered from poultry revealed wider range of resistance to ciprofloxacin (70-88%), …………………… 93.8 million foodborne illnesses and 155,000 fatalities annually worldwide [46].” The authors only compared the AST results of other with this study but did not explained why there is a difference. The discussion part need extensive revision otherwise it is not suitable to publish.
Response: we extensively revised entire manuscript and the sentences and make it clear for the reader.
Materials and Methods
- Page 9. Section 4.2. Sampling design: Please explain the samples details. How and from where the meat and milk samples were collected? From local market or from slaughterhouses?
- Page 9. Section 4.3. 4.3. Conventional culture method: “Briefly, each of the samples was inoculated in nutrient broth and incu[1]bated at 37°C for 12 hours for initial growth of Escherichia coli, Salmonella and Staphylococ[1]cus aureus. Briefly, 0.5 gram of each sample was inoculated in nutrient broth and incubated at 37°C for 12 hours for initial growth of Escherichia coli, Salmonella and Staphylococcus…..” These two sentences are repeated twice
Response: deleted.
- Page 10. Section 4.3. 4.3. Conventional culture method: “For the isolation of Campylobac[1]ter, all samples were directly inoculated on selective campylobacter base agar (Oxoid Ltd, UK) containing antibiotics and 5-7% sheep blood.” Which antibiotics ??????
Response: we added the information with reference.
- Page 10. Section 4.4. Molecular detection: Better to change chromosomal DNA into "Whole genomic DNA"
Response: replaced with "Whole genomic DNA"
- Page 10. Section 4.4. Molecular detection: “According to our previous studies [42], we found boiling method is effec[1]tive well-established method for chromosomal DNA extraction as teratogenic technique instead of using commercial kit.” Delete this sentence
Response: deleted
- Page 10. Section 4.5. Determination of phenotypic resistance pattern” “The zone of inhibition adjacent to the disks was measured using conventional methods and compared to the break points of CLSI.” which conventional methods?
Response: we revised the sentences and clearified
- Page 11. Section 4.5. Determination of phenotypic resistance pattern: “Based on our filed survey [23],” Field or Filed ????????
Response: sorry for our careless mistake. And was corrected
- Page 11. Section 4.5. Determination of phenotypic resistance pattern: “Multidrug resistance (MDR) bacteria are defined using the criteria as established by Magiorakos et al. [84]” Rephrase this sentence
Response: we revised the sentence
- Page 11. Section 4.5. Determination of phenotypic resistance pattern: “The Escherichia coli ATCC 25922 strain was used as known positive control.” Rephrase this sentence
Response: we revised the sentence
- Page 11. Section 4.6. Determination of genotypic resistance pattern: Why the authors detect ARGs of few drugs only? Why not they include Carbapenem, Tigecycline, and Colistin related genes? This is the major gap in your study that you missed the important drugs prevalence which is very common in animals and humans and all these drugs are the last resorts for AMR.
Response: For your kind information Carbapenem, Tigecycline, and Colistin are banded in our country and are not used in our field condition, therefore, based on our previous study results (Ref. no. 30) we did the common antibiotics used in the study area.
- Page 11. Section 4.6. Determination of genotypic resistance pattern: “Extended Spectrum β-Lactamases (ESBL) genes (blaTEM, blaSHV, and blaCMY)” Where is blaCTX-M which is the main ESBL gene? Why you only choose just blaTEM, SHV and CMY? Again here you missed the most important and common ESBL gene blaCTX-M.
Response: This is our study limitation and were explained in the last para of discussion section.
- Table 4: Would be better to move this table to supplementary as well the references of this table.
Response: we again finally appreciate your kind effort to give comment for the improvement of our manuscript and will be move to supplementary file.

Reviewer 2 Report
Thanks for effort in the work and kind presentation.
However, certain points could be considered during revision.
Generally, some spelling mistakes are present, so, the manuscript should be revised again.
Abstract:
- Any abbreviation should be written firstly as full words followed by the abbreviation (such as E. coli), then the abbreviation is written all over the manuscript.
- And not &.
The key words, if possible, differ from words in the title.
Introduction:
- The prime cause of foodborne infections is the presence of bacterial ???, please, complete the sentence.
- L. monocytogenes, and E. coli (Listeria monocytogenes and Escherichia coli).
- Please, identify the species of Salmonella causing food poisoning (S. paratyphoid species), many some species as S. pullorum not a food borne pathogen.
- Animal derived food products most commonly red meat, white meat ?? are the important vehicles through which people may expose with foodborne pathogens especially bacteria. Rephrasing, please.
- In Bangladesh, no comprehensive research work was carried out on the determination of antimicrobial drug susceptibility profile of common foodborne pathogens recovered from livestock and poultry food products throughout the country. How?. You cited ref. [37] Parvin, M.S., Talukder, S., Ali, M.Y., Chowdhury, E.H., Rahman, M.T., Islam, M.T., 2020. Antimicrobial Resistance Pattern of Escherichia coli Isolated from Frozen Chicken Meat in Bangladesh. Pathog. . https://doi.org/10.3390/pathogens9060420
Please, make another search about this point in your country and include references.
Bacteriological assessments of foodborne pathogens in poultry meat at different super shops in Dhaka, Bangladesh, 2019. https://doi.org/10.4081%2Fijfs.2019.6720
Occurrence of Campylobacter spp. in Selected Small Scale Commercial Broiler Farms of Bangladesh Related to Good Farm Practices, 2020. https://doi.org/10.3390%2Fmicroorganisms8111778
Outbreak of Salmonella in poultry of Bangladesh and possible remedy, 2019. https://doi.org/10.5455/jabet.2019.d30
Discussion:
- The presence of antibiotic-resistant foodborne pathogens in food may lead to gastro-intestinal disorders in human beings. Please, add references.
- What is the species of Salmonella and Camplybacter that were identified?
Materials and Methods:
- Are tissue samples (poultry and livestock) were taken at processing?
References:
Please, check the references again.
- 2. Haile, A.F., Kebede, D., Wubshet, A.K., 2017. isolated from bovine in Jimma , Ethiopia : abattoir- based survey. Ethiop. Vet. J. 21, 109–120. In complete reference.
Best wishes
Author Response
Reviewer # 2
Thanks for effort in the work and kind presentation.
However, certain points could be considered during revision.
Generally, some spelling mistakes are present, so, the manuscript should be revised again.
Abstract:
- Any abbreviation should be written firstly as full words followed by the abbreviation (such as E. coli), then the abbreviation is written all over the manuscript.
Response: we did it
- And not &. The key words, if possible, differ from words in the title.
Response: we revised it
Introduction:
- The prime cause of foodborne infections is the presence of bacterial ???, please, complete the sentence.
Response: sorry for our careless mistake and was revised
- monocytogenes, and E. coli (Listeria monocytogenes and Escherichia coli).
Response: revised and added
- Please, identify the species of Salmonella causing food poisoning (S. paratyphoid species), many some species as S. pullorum not a food borne pathogen.
Response: revised and added
- Animal derived food products most commonly red meat, white meat ?? are the important vehicles through which people may expose with foodborne pathogens especially bacteria. Rephrasing, please.
Response: we revised the sentence
- In Bangladesh, no comprehensive research work was carried out on the determination of antimicrobial drug susceptibility profile of common foodborne pathogens recovered from livestock and poultry food products throughout the country. How?. You cited ref. [37] Parvin, M.S., Talukder, S., Ali, M.Y., Chowdhury, E.H., Rahman, M.T., Islam, M.T., 2020. Antimicrobial Resistance Pattern of Escherichia coli Isolated from Frozen Chicken Meat in Bangladesh. Pathog. . https://doi.org/10.3390/pathogens9060420
Please, make another search about this point in your country and include references.
Bacteriological assessments of foodborne pathogens in poultry meat at different super shops in Dhaka, Bangladesh, 2019. https://doi.org/10.4081%2Fijfs.2019.6720
Occurrence of Campylobacter spp. in Selected Small Scale Commercial Broiler Farms of Bangladesh Related to Good Farm Practices, 2020. https://doi.org/10.3390%2Fmicroorganisms8111778
Response: we revised the statement and added the above mentioned information including references
Discussion:
- The presence of antibiotic-resistant foodborne pathogens in food may lead to gastro-intestinal disorders in human beings. Please, add references.
Response: Reference added
- What is the species of Salmonella and Camplybacter that were identified?
Response: Salmonella and Camplybacter Spp were identified by using the specific perimeters mentioned in methods section
Materials and Methods:
- Are tissue samples (poultry and livestock) were taken at processing?
Response: no, we clarified the collection of samples in the methodology section and revised the sentences
References:
Please, check the references again.
- isolated from bovine in Jimma , Ethiopia : abattoir- based survey. In complete reference.
Response: Sorry for our careless mistake and complete information was added in reference section.

Round 2
Reviewer 1 Report
Thanks for revising the manuscript as per suggestion. The manuscript seems better than before however there are some few more suggestions need to be addressed.
Results
1. Section 2.1. Prevalence of AMR pathogens in animal derived food products: "E. coli, Salmonella, Staphylococcus, and Streptococcus species were isolated from livetstock origin food product & by-product samples and, E. coli, Salmonella, and Staphylococcus species were isolated from poultry origin food samples throughout the country." I think the authors did not understand my last suggestion. Merging these two sentences means that no need to repeat the bacterial names twice. Just make one sentence. Please revise it.
2. 2.2. Phenotypic resistance pattern: "The AST result of Salmonella showed highest resistance to penicillin (96.15%) followed by ampicillin (91.48%), oxytetracycline (OT) (82.2%), amoxicillin (73.1%), cloxacillin (67.85%) whilst highest susceptible was recorded to gentamicin (CN) (82.91%) followed by ceftriaxone (CRO) (58.88%)..." Here the authors used abbreviations for some antibiotics while some dont have. Please revise it and write the abbreviations accordingly to the previous papers published. For example, you used (CN) for gentamicin while in previous papers GEN is commonly used. The authors need to revise it throughout the manuscript.
Discussion
3. "The overuse and misuse or unnecessary use of antibiotics in livestock has become a great concern for public health regime." I suggest to write the overuse and misuse only. no need to write unnecessary use. revise the sentence. The word "regime" is not suitable here, use any other alternative word.
Author Response
Results
- Section 2.1. Prevalence of AMR pathogens in animal derived food products:"E. coli, Salmonella, Staphylococcus, and Streptococcus species were isolated from livetstock origin food product & by-product samples and, E. coli, Salmonella, and Staphylococcus species were isolated from poultry origin food samples throughout the country." I think the authors did not understand my last suggestion. Merging these two sentences means that no need to repeat the bacterial names twice. Just make one sentence. Please revise it.
Response: Sorry for misunderstanding, we revised the text deleted the repetition of the bacterial names twice.
- 2.2. Phenotypic resistance pattern: "The AST result of Salmonella showed highest resistance to penicillin (96.15%) followed by ampicillin (91.48%), oxytetracycline (OT) (82.2%), amoxicillin (73.1%), cloxacillin (67.85%) whilst highest susceptible was recorded to gentamicin (CN) (82.91%) followed by ceftriaxone (CRO) (58.88%)..." Here the authors used abbreviations for some antibiotics while some dont have. Please revise it and write the abbreviations accordingly to the previous papers published. For example, you used (CN) for gentamicin while in previous papers GEN is commonly used. The authors need to revise it throughout the manuscript.
Response: Based on Hi Media, India company abbreviations we revised the entire text both in result and method sections as well as figures as follows
A number of 16 different antimicrobial disks (Hi media, India) were used for AST of all four foodborne pathogens such as penicillin (P, 10 units), ampicillin (AMP, 10 μg), amoxicillin (AMX, 30 μg), cloaxcillin (COX, 5 μg), ceftriaxone (CTR, 30 μg), tetracycline (TE, 30 μg), doxycycline (DO, 30 μg), oxytetracycline (O, 30 μg), sulfamethoxazole-trimethoprim (COT, 25 μg), gentamicin (GEN, 10 μg), erythromycin (E, 15 μg), ciprofloxacin (CIP, 5 μg), streptomycin (S, 10 μg), levofloxacin (LE, 5 μg), enrofloxacin (EX, 5 μg), and neomycin (N, 30 µg).
Discussion
- "The overuse and misuse or unnecessary use of antibiotics in livestock has become a great concern for public health regime."I suggest to write the overuse and misuse only. no need to write unnecessary use. revise the sentence. The word "regime" is not suitable here, use any other alternative word.
Response: Sorry for misunderstanding, we revised the sentence and replaced the word “regime” as “authorities”